# Workplace Health Promotion Embedded in Medical Surveillance: The Italian Way to Total Worker Health Program

**DOI:** 10.3390/ijerph20043659

**Published:** 2023-02-18

**Authors:** Nicola Magnavita

**Affiliations:** 1Post-Graduate School of Occupational Health, Università Cattolica del Sacro Cuore, 00168 Rome, Italy; nicolamagnavita@gmail.com; Tel.: +39-347-330-0367; 2Department of Woman, Child and Public Health, Fondazione Policlinico Universitario Agostino Gemelli IRCCS, 00168 Rome, Italy

**Keywords:** occupational health, policy, practice, sustainability, participatory approach, salutogenesis, effectiveness, well-being

## Abstract

In 2011, NIOSH launched the Total Worker Health (TWH) strategy based on integrating prevention and health promotion in the workplace. For several years now, in Italy, this integration has led to the creation of workplace health promotion embedded in medical surveillance (WHPEMS). WHPEMS projects, which are also implemented in small companies, focus each year on a new topic that emerges from the needs of workers. During their regular medical check-up in the workplace, workers are invited to fill in a questionnaire regarding the project topic, its outcome, and some related factors. Workers receive advice on how to improve their lifestyles and are referred to the National Health Service for any necessary tests or treatments. Results collected over the past 12 years from more than 20,000 participants demonstrate that WHPEMS projects are economical, sustainable, and effective. The creation of a network of occupational physicians who are involved in WHPEMS projects could help to improve the work culture, health, and safety of workers.

## 1. Introduction

The Total Worker Health (TWH) Program, defined “as policies, programs, and practices that integrate protection from work-related safety and health hazards with promotion of injury and illness-prevention efforts to advance worker well-being” [1], was launched by NIOSH in 2011 as a new public health strategy that quickly gained support in several countries. The TWH approach provides a comprehensive framework for enhancing worker well-being, health, and safety. The general well-being of employees is improved by adopting an integrated approach that prioritizes safety and simultaneously undertakes other workplace initiatives (such as healthy work design, employee training and development, accident and sickness prevention efforts, etc.). The principles of TWH, summarized in a handbook published in 2016 [2], provide an up-to-date preventive strategy that is compatible with traditional occupational safety and health prevention procedures but also acknowledges the potential importance of job-related issues for the health of employees, their families, and their communities.

In order to promulgate this strategy, NIOSH has set up Centers of Excellence for Total Worker Health to provide the scientific data required to create new solutions for TWH challenges. The research carried out at these centers produces new information and proposes creative solutions to problems currently encountered by modern enterprises. Through nonfunded collaborations with governmental and nonprofit organizations, NIOSH has also established a TWH partnership program that aims to increase the visibility and impact of TWH initiatives by publicly acknowledging the efforts companies make in order to advance THW research and practice.

The development and application of TWH principles have been the subject of considerable debate among prevention operators since the new approach can lead to ethical dilemmas. Many believe that decision-making should be based on ethical constructs [3], but the new model has often been seen to be implemented amid a climate of corporate supremacy and in pursuit of a relentless neoliberal agenda. As a result, integration attempts have sometimes failed to effectively combine wellness with safety and health and have instead placed a focus on individual worker responsibility for health [4]. Moreover, there is no agreement on which program components should be considered “TWH,” how much emphasis should be placed on organizational work settings, or which organizational or individual outcomes are the most important [5]. To clarify these fundamental points, four factors have been proposed for consideration: (i) coordination and interaction of workplace programs across domains; (ii) assessment of both work and nonwork exposures; (iii) emphasis on interventions to make the workplace more health-promoting; and (iv) essential participation of workers in prioritizing and planning intervention to foster self-efficacy. Basically, TWH involves organizational change designed not only to incorporate two managerial functions with specific objectives, legal obligations, and internal incentives and resources but also to steer the firm toward salutogenesis [5].

Furthermore, there is little concrete evidence of the obstacles and enabling factors involved in the acceptance, implementation, and sustained maintenance of TWH initiatives [6]. Methods such as ROI (return on investment), which are used to assess the effectiveness of programs, need to be carefully evaluated since their application has led to substantial criticism [7]. In particular, a thorough study has not been made regarding the effectiveness of TWH interventions in small businesses that require outside help to expand or enhance current workplace health and safety initiatives [8]. As suggested by Rohlman et al. [9], at least eight key points should be considered: value and return on investment (ROI), organizational factors, program design, employee engagement, low-cost strategies, evaluation, and integration.

The US initiative has aroused interest in several countries that have tried to adapt the philosophy of TWH to different national conditions. For example, Germany has adopted “workplace health management,” a comprehensive approach to safeguard, advance, and manage employees’ health at work [10]. The models followed by German companies include four subcategories of intervention: “occupational health and safety” and “reintegration management” include legally required procedures, while “workplace health promotion” and “personnel development” share similarities with TWH but are tailored to meet specific company requirements.

In Italy, the Ministry of Health, through the National Prevention Plan (NPP) for the 5-year period 2020–2025 [11], has made explicit reference to the TWH approach in the Central Support Line No. 3, “Activation of technical tables for the strengthening of the overall health of the worker according to the Total Worker Health approach.” On 15 June 2022, the Italian Society of Occupational Medicine (SIML) approved the creation of a working group on health promotion that would assist occupational physicians in creating workplace health promotion initiatives in keeping with the TWH approach and would establish a link between general practitioners (GPs) and occupational physicians (OPs) [12].

While there is little doubt that health promotion is part of the occupational health mission and that OPs and GPs share the same public health goals, collaboration between these different professional categories is far from automatic and requires programs, resources, and guidelines. Reports of experiences in the USA showed that resource shortages, the organizational structure of safety and health services, and incompatible techniques were obstacles to an OP–GP partnership, whereas knowledge of TWH methodologies, proximity to TWH Centers of Excellence, and leadership initiatives acted as facilitators [13].

However, a transfer of the US experience to other social contexts must take into consideration differences in health and social care systems. In the USA, healthcare is predominantly private. Companies provide resources for employee healthcare through insurance policies; consequently, they have a vested interest in reducing disease among their employees in order to cut insurance costs. On the contrary, the Italian National Health Service (NHS) guarantees free healthcare for all, and workers’ insurance against accidents and occupational diseases is compulsory. Companies contribute through taxation to the NHS and the National Institute for Social Security (INPS) and pay compulsory insurance to the National Institute of Insurance for Accidents and Occupational Diseases (INAIL). Therefore, they do not have an immediate economic return from a reduction in injuries or illnesses. Furthermore, in Italy, over 92% of active enterprises are small or very small and employ 82% of workers [14]. These conditions underlie the great difficulty in finding resources to support health promotion programs, particularly in small and medium-sized Italian businesses. On the other hand, Italy has numerous occupational physicians (around 5500) who annually take care of 14 million workers by carrying out approximately 10 million examinations per year [15]. This situation increases the possibility of conducting health promotion programs in the workplace.

For many years, our university has developed techniques that enable health promotion programs to be disseminated during health surveillance activities in the workplace. This article describes the workplace health promotion method embedded in medical surveillance (WHPEMS), which has been applied by occupational physicians, also in small companies. By exploiting the unique characteristics of the Italian NHS that offers free and universal access, this method has limited the need for resources and has been able to envisage and implement the integration criteria of prevention and promotion postulated by the NIOSH. The purpose of this article is to provide OPs with a simple, effective, and economical method of promoting health in the workplace. The manuscript aims to present in detail an initiative in the methodology applied in occupational settings.

## 2. Materials and Methods

The essential prerequisite for carrying out successful workplace initiatives is the management of occupational risks. In order to implement the integrative approach to the concept of occupational health, which is the basis of the TWH, it is essential, first of all, to provide for the prevention of occupational risks. Only when this is achieved is it legitimate to plan the promotion of workers’ health. Our school adopted the A.S.I.A. (assessment, surveillance, information, audit) model [16] for risk management., According to this model, the various phases of risk assessment, surveillance of workplaces and workers, and occupational information and training must be closely linked and shared among all prevention operators. The identification of crucial aspects in the system must result in investigations or audits designed to suggest action for improving the work environment. In small companies, in particular, risk management requires a significant contribution on the part of the occupational physician, who participates personally in the various phases of the process. In fact, the intervention of an occupational physician in the workplace is essential for identifying the workers’ health needs.

WHPEMS interventions are conducted every year in accordance with a proposal based on indications previously provided by the workers. The occupational physician collects this information during visits to the workplace when he/she invites workers to describe their work cycle, identify any critical issues, and by means of participatory ergonomic groups (in Italian, Gruppi di Ergonomia Partecipativa, G.E.P.), make proposals for workplace improvement [17]. The G.E.P. technique, which is taught by our school and is freely accessible, consists of working with small groups of workers to improve the working environment, processes, and procedures. The groups are made up of all workers who contribute to supplying a product, for example, in a hospital ward, from doctors, nurses, and ancillary personnel, and at a petrol station by the owner, yard workers and cashiers, and so on. All contribute to describing their work and identifying critical points in relation to each for which they suggest possible solutions. The simplest and cheapest shared solutions are proposed by the doctor to the company. The interviews with the workers also serve the doctor to identify the topics of the promotion campaigns.

Over the years, chosen topics have concerned pathologies with a particular impact on productivity and determinants of well-being. Consequently, headache disorders (one of the main causes of years lived with disability globally) [18], musculoskeletal disorders [19], and syncope (responsible for a significant increase in the risk of occupational accidents and termination of employment) [20] have been investigated. Symptoms associated with air quality [21,22,23] and with the low quality of work organization [24] have also been studied due to their high prevalence. In other cases, the search for strategies to increase work engagement and reduce burnout and occupational stress [25,26] has been the main objective. The close relationship between occupational stress and sleep disturbances [27] and between workplace violence and stress [28] has led to a project designed to promote sleep quality [29] and another for encouraging nonviolent behaviors [30]. Aging and its consequent effect on working attitudes [31,32] prompted another project, supported by a comparison with European experiences in the framework of an international collaboration [33,34,35]. The pandemic caused by the SARS-CoV-2 virus led us to evaluate the onset of post-COVID-19 syndrome in workers [36,37] and seek ways of promoting their recovery. In 2022, we addressed the issue of eating disorders [38,39], and at the request of workers, in 2023, we will develop the logical continuation of this project with a program aimed at encouraging the spread of the Mediterranean diet [40].

Workers are invited to participate in WHPEMS action during their regular medical examination in the workplace. There is no sampling because all those undergoing health surveillance are invited to participate. Workers are free to accept or refuse, but the vast majority agree to provide the requested data and are eligible for the promotion. Those who agree to participate sign a consent form and receive a questionnaire made up of three parts: the first investigates the topic of the promotion action; the second measures possible outcomes; while the third measures possible confounders, mediators, and modulators of the relationship observed. For example, since a consistent amount of scientific evidence indicates that increased adherence to the Mediterranean diet is associated with favorable mental and physical health outcomes [41,42], the 2023 program will aim to promote the Mediterranean diet. The questionnaire, which can be completed on paper or online, is made up of questions that refer to the three aforementioned areas: (i) a 14-point Mediterranean Diet Adherence Screener (MEDAS) [43] to evaluate the degree of adherence to the Mediterranean diet and provide advice for the workers; (ii) an analysis of lifestyles, metabolic parameters, and mental health with the General Health Questionnaire [44] as outcomes; and (iii) an evaluation of occupational stress using Siegrist’s effort/reward imbalance model [45,46], of sleep disorders with the Sleep Condition Indicator [47], and of trauma with the Violent Incident Form [48] as possible cofactors in the relationship. The questionnaire form can be provided upon request to doctors who intend to apply this program in their companies. Workers’ responses can also be collected online. No charge is required for these services, which are aimed at improving health in the workplace and also in other companies.

Workers receive immediate health promotion advice from the occupational physician and guidance on where to access NHS facilities for diagnostic tests or necessary treatments. After processing the data contained in the questionnaires, the occupational physician transmits the results in a collective anonymous form to the employer, the protection service manager, and the workers’ safety representatives so that eventual collective promotion measures can be adopted.

## 3. Results

Over the past 12 years, the occupational physicians in our occupational health unit have conducted the WHPEMS programs reported in Table 1. More than 20,000 workers have participated, and over 1000 of them have contacted their GPs or other NHS facilities.

Although it was not obligatory to join the programs, most workers participated. In cases where screening revealed a disease, the medical examinations performed immediately after completing the questionnaire were used to expand the workers’ medical history, investigate comorbidities, and evaluate any alterations. Workers were invited to start or continue their diagnostic/therapeutic pathway at the NHS. In these cases, the OP took the opportunity to contact the worker’s GP, to whom he/she sent a letter via the worker, indicating the data emerging from the examination and the patient’s possible needs. The worker was invited to provide information on the evolution of his/her pathology that would, however, be further checked by the OP during the worker’s next routine examination. In most cases, when OPs detected only incorrect habits or risky behaviors, contact with the worker was used to reinforce the salutogenic approach by pointing out the advantages of correct lifestyles.

The data collected through questionnaires were processed electronically. The results of each survey were reported to the companies, the corporate prevention service, and the workers’ safety representatives in order to contribute to the growth of the work culture. Besides offering workers advice and providing companies with useful indications, WHPEMS activities have enabled researchers to produce some scientific publications [49,50,51,52,53,54,55,56,57,58,59,60,61,62,63]. Furthermore, by annually reiterating surveys on the same cohorts of workers, it has been possible to carry out longitudinal studies to clarify the causal link between exposure to risk and damage to health [64,65].

## 4. Discussion

The experiences conducted by our occupational medicine unit over a period of several years and summarized in this paper indicate that including health promotion in prevention activities required by law presents characteristics worthy of attention. First of all, since this method makes use of an existing health and safety service, it is very economical because it does not require companies to make a significant commitment of economic resources. Furthermore, it does not interfere with the work of OPs. On the contrary, it guarantees an important supply of data for analyzing workers’ health conditions. Usually, a lack of resources is reported to be the most common obstacle to WHP programs, whereas strong management support is held to be the most common facilitator of this type of intervention [66]. The modest quantity of resources required by this method makes it possible to apply WHPEMS projects even in small businesses. The well-being of workers is of great interest to companies because research data demonstrate that health promotion initiatives that also focus on the physical work environment and organizational structure of the workplace can significantly influence job-related outcomes, including absenteeism [67]. The best method of freeing resources for health promotion is to raise the standard of occupational health at work. In fact, the true goal of occupational medicine is to improve workplace health. This can be accomplished more effectively by adopting a broad strategy that considers occupational risks, technical and medical knowledge, ergonomic workplace modifications, and behaviors and lifestyles that may encourage the development of diseases and consequently limit working capacity. One of the most important characteristics of projects is continuity: frequently, a project can only be sustained if it helps to recover resources or significantly increases production. Many ventures fail if they are not consistently financed, although there are a few exceptions to this general norm. In the Netherlands, some projects continued after national funds had run out because companies recognized their value and decided to continue them by financing the experience themselves [68]. In Italy, an incentive for the implementation of health promotion programs is represented by the fact that companies that carry out interventions supplementary to the obligatory prevention of occupational damage can request a reduction in the insurance premiums they have to pay to the National Institute (INAIL). This immediate economic benefit largely exceeds the very modest expense of WHPEMS programs and ensures that any other benefit of the intervention that should occur in the future due to reduced absenteeism, increased productivity, etc., will be a net gain. Furthermore, companies derive image benefits from WHPEMS programs. For example, one of the companies in which we carried out the interventions was awarded for best practices in the 2007 European campaign “Lighten the load” and in the 2016/2017 European campaign “Safer and healthier work at any age—occupational safety and health in the context of an aging workforce.” A second interesting characteristic of this method is its sustainability, demonstrated over a period of more than 12 years. Promotion continuity depends on two factors: corporate social responsibility and the motivation of workers. WHPEMS interventions give companies a tangible demonstration of their social responsibility. It is important to encourage corporate social responsibility because workplace health promotion and company social responsibility are related. They have mutually beneficial effects based on leadership that respects autonomy and voluntary participation accompanied by recognition of specific goals that comply with the parameters of company sustainability policies [69]. Worker participation is another factor related to sustainability. The decision to choose “positive” salutogenic objectives that change from year to year is designed to increase motivation, which is difficult to achieve if the objective is always the same and of a negative nature, e.g., avoiding drinking, smoking, taking drugs, etc. The salutogenic approach plays a significant role in research and practice related to public health and health promotion. This approach might help to solve some of today’s most pressing public health issues (e.g., the promotion of mental health) and might produce a sound theoretical foundation for health promotion [70,71].

The third aspect is the participatory nature of WHPEMS. Workers contribute by proposing topics that become the subject of promotion projects and also by responding to their doctor’s advice. Research has shown that the participatory approach is a good example to follow for promoting health [72]. Choosing topics shared with workers ensures that program goals are relevant to workers’ health and that they will derive the greatest benefit for their own health.

A further interesting characteristic of WHPEMS projects is that they enable researchers to systematically collect interesting occupational medicine variables, such as perceived occupational stress and workplace violence. The latter is considerably underreported in workers’ spontaneous reports [73]; however, a survey based on questionnaires makes it possible to collect experiences in a systematic way. Information on violence experienced and perceived stress is highly sensitive material; the fact that it is collected as a collateral aspect of an investigation targeting other clinical problems may reduce the risk of overreporting, a phenomenon often linked to compensation expectations [74,75,76]. The surveys we conducted made it possible to investigate very delicate aspects such as occupational stress and violence in the workplace. As discussed above, the implementation of WHPEMS interventions is an advantage for companies and workers. It is also a great advantage for the occupational doctor, who has the possibility of improving the health of workers and earning an optimal relationship with them. It is useful to remember that the occupational doctor is not chosen by the workers and does not enjoy the fundamental doctor–patient relationship on which medicine is based. By implementing WHPEMS programs and improving working life and work culture, he/she can gain the trust and cooperation of workers.

Lastly, by creating a real flow of information between OPs and GPs, the WHPEMS method avoids duplicating activities and wasting resources during public health action. In a health system in which resources are limited, waste is ethically unacceptable. Regular mandatory health surveillance in the workplace makes it possible to evaluate the effectiveness of promotional interventions. Numerous requests have been made to overcome obstacles such as organizational, interpersonal, and structural barriers that can hamper cooperation between OPs and GPs, and suggestions have been put forward in order to successfully achieve this aim [77].

The recent COVID-19 pandemic has shown how important the relationship between OPs and GPs is in disseminating and sustaining good practices. It has been observed that when public health shocks occur, policymakers can encourage pertinent learning processes by supporting knowledge and education to raise people’s understanding of preventive health practices [78]. A change in living behavior may have resulted from the lockdown, during which residents were advised to spend as little time outside their homes as possible and work from home. This life-changing event may have altered lifestyle choices, which have a crucial role in the development and progression of illnesses [79]. By implementing health promotion strategies, communities may be better able to avoid, identify, and contain epidemic threats. They may also improve the effective allocation of scarce resources to high-impact public health systems [80]. This could be of the utmost importance for dealing with any other emergencies in the future. For example, in the case of post-COVID-19 syndrome, maintaining or regaining post-COVID-19 workability might reasonably follow a typical biopsychosocial framework enhanced to account for the cyclical nature of the symptoms. This should include adaptable, ongoing, longer-term return-to-work planning that addresses several levels of workability barriers, produced jointly by employees and line managers with assistance from OPs, GPs, and an improved organizational culture [81].

A limitation of the advantageous WHPEMS method is the considerable effort needed to plan the annual project and carry out a statistical analysis of the data collected. To overcome this limitation, the occupational medicine unit of the Catholic University of the Sacred Heart makes its project available to all OPs and provides free data processing to interested researchers. The aim of this work was to introduce a method for carrying out health promotion interventions that would be accessible to OPs, even in small companies. Despite the numerous actions undertaken worldwide, there is still little and inconsistent evidence regarding the effectiveness of strategies for improving the implementation of health-promoting policies and practices in the workplace [82]. We are convinced that a simple, cheap, and effective method such as the one proposed can significantly improve occupational health promotion actions.

## 5. Conclusions

The aforementioned WHPEMS method, based on continuous health promotion within regular health surveillance activities conducted in the workplace, offers many advantages such as cost-effectiveness, sustainability, and a participatory approach, all of which recommend extensive application. Moreover, it is a valid tool for collecting useful information for the surveillance of workers and for improving work environments. It also provides a solid foundation for creating a two-way information flow between occupational physicians and NHS doctors. The critical issues inherent in this method, as in all health promotion programs, concern the need to analyze evidence, design the survey correctly, and process the results. Our university gives occupational physicians access to experiences gained in over 12 years of promotion activities and aims to create a network for all those interested in promoting health in the workplace.

## 6. Patents

The A.S.I.A. method for risk management and the Participatory Ergonomics Groups (G.E.P.) method are registered trademarks of the author and are freely accessible by all prevention operators who intend to apply them in the workplace.

## Figures and Tables

**Table 1 ijerph-20-03659-t001:** Health promotion programs implemented in the last 12 years by the occupational medicine unit of the Catholic University of the Sacred Heart.

Year	Topic	References
2022	Eating behavior disorders	-
2021	Post-COVID syndrome	[49] ^1^
2020	Syncope and presyncope	[50,51]
2019	Headache	[52,53,54]
2018	Musculoskeletal disorders	-
2017	Work engagement	[55]
2016	Sleep health promotion	[56]
2015	Aging and ability to work	[57]
2014	Violence at work	[58]
2013	Work organization	[59,60]
2011–2012	Indoor air quality	[61,62,63]

^1^ Submitted.

## Data Availability

The data collected in the studies that have been the subject of publications are contained in publicly accessible data repositories, which are indicated in the respective articles.

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
