# Peer review of "Workplace Health Promotion Embedded in Medical Surveillance: The Italian Way to Total Worker Health Program"

_ijerph, 2023, doi:10.3390/ijerph20043659_

Round 1

Reviewer 1 Report

The subject paper is quite interesting and well written, aimed to explain potential applications of TWH Program in the author's country.

The author declares no conflict of interest, but he added the section "Patent" reporting two owned patents cited in the text. Please revise it.

I suggest to extend the discussion about COVID-19 and its sequelae, also regarding the link with the diet. Feel free to discuss the following paper:

doi: 10.1016/j.foodpol.2022.102365

doi: 10.1186/s12889-021-11264-z

doi: 10.34172/ijoem.2020.1984

doi: 10.1136/bmjgh-2021-007365

doi: 10.1093/occmed/kqac086

At last, I suggest to add the term "Program" at the end of the title, and revise the text for typos and avoiding the use of special carachters (i.e., &).

Author Response

Reviewer #1

The subject paper is quite interesting and well written, aimed to explain potential applications of TWH Program in the author's country.

The author declares no conflict of interest, but he added the section "Patent" reporting two owned patents cited in the text. Please revise it.

Response: I thought it appropriate, also in the light of what the reviewer asked, to explain in the text that the ASIA method for risk management and the GEP technique for worker participation in job improvement are techniques taught by our school and freely accessible.

I suggest to extend the discussion about COVID-19 and its sequelae, also regarding the link with the diet. Feel free to discuss the following paper:

doi: 10.1016/j.foodpol.2022.102365

Gao Y, Lopez RA, Liao R, Liu X. Public health shocks, learning and diet improvement. Food Policy. 2022 Oct;112:102365. doi: 10.1016/j.foodpol.2022.102365.

doi: 10.1186/s12889-021-11264-z

van der Werf ET, Busch M, Jong MC, Hoenders HJR. Lifestyle changes during the first wave of the COVID-19 pandemic: a cross-sectional survey in the Netherlands. BMC Public Health. 2021 Jun 25;21(1):1226. doi: 10.1186/s12889-021-11264-z.

doi: 10.34172/ijoem.2020.1984

not found

doi: 10.1136/bmjgh-2021-007365

Zhao F, Bali S, Kovacevic R, Weintraub J. A three-layer system to win the war against COVID-19 and invest in health systems of the future. BMJ Glob Health. 2021 Dec;6(12):e007365. doi: 10.1136/bmjgh-2021-007365.

doi: 10.1093/occmed/kqac086

Lunt J, Hemming S, Burton K, Elander J, Baraniak A. What workers can tell us about post-COVID workability. Occup Med (Lond). 2022 Aug 15:kqac086. doi: 10.1093/occmed/kqac086.

Response: I welcomed the suggestion to add some comments on health promotion conducted in relation to the COVID-19 pandemic that I had not covered extensively because the related findings are still largely unpublished. I have added the following paragraph: “The recent Covid-19 pandemic has shown how important the relationship between POs and GPs is in disseminating and sustaining good practices. It has been observed that, when public health shocks occur, policymakers can encourage pertinent learning processes by supporting knowledge and education to raise people's understanding of preventative health practices. [Gao] A change in living behavior may have resulted from the lockdown, during which residents were advised to spend as little time outside of their homes and work from home [van der Werf]. By implementing health promotion strategies, countries may be better able to avoid, identify, and contain epidemic threats. They may also improve the effective allocation of scarce resources to high-impact public health systems [Zhao]. For example, in the case of post-Covid syndrome, maintaining or regaining post-Covid workability might reasonably follow a typical biopsychosocial framework enhanced to account for the cyclical nature of the symptoms. This should include adaptable, ongoing, longer-term return-to-work planning that addresses several levels of workability barriers, produced jointly by employees and line managers with assistance from OPs, GPs, and an improved organizational culture [Lunt]”

At last, I suggest to add the term "Program" at the end of the title, and revise the text for typos and avoiding the use of special carachters (i.e., &).

Response: I gladly accepted the suggestion to change the title and corrected some typos in the text

Reviewer 2 Report

Dear Author

I carefully evaluated your manuscript, finding it very interesting and well written.

No changes are required.

I suggest acceptance in the present form

Best Regards

Author Response

Reviewer #2

Dear Author

I carefully evaluated your manuscript, finding it very interesting and well written.

No changes are required.

I suggest acceptance in the present form

Best Regards

Response: I sincerely thank the reviewer for appreciating my work.

Reviewer 3 Report

The manuscript addresses an interesting topic and presents a novel action to improve the quality of life of health professionals. The approach to the concept of health is integrative. It would be advisable to make it clear that the manuscript in this case aims to present in detail an initiative in methodology applied in healthcare settings. By not providing specific data obtained after its implementation, it would be necessary to review the sections that structure the manuscript. The procedures for contacting the participants and selecting the sample must be specified in the manuscript. Likewise, any reference to the sociodemographic characteristics of the participants would also be interesting, if changes have been observed over the years of program implementation. The results obtained could be summarized in some way in the manuscript, to check if there have been differences in relation to the different periods of implementation. Highlight the practical implications of this methodology, benefits for the professionals themselves and also the advantages for the institution.

It would be interesting if the author could provide some specific data on the results obtained in the implementation of this initiative.

Author Response

Reviewer #3

The manuscript addresses an interesting topic and presents a novel action to improve the quality of life of health professionals. The approach to the concept of health is integrative. It would be advisable to make it clear that the manuscript in this case aims to present in detail an initiative in methodology applied in healthcare settings. By not providing specific data obtained after its implementation, it would be necessary to review the sections that structure the manuscript. The procedures for contacting the participants and selecting the sample must be specified in the manuscript. Likewise, any reference to the sociodemographic characteristics of the participants would also be interesting, if changes have been observed over the years of program implementation. The results obtained could be summarized in some way in the manuscript, to check if there have been differences in relation to the different periods of implementation. Highlight the practical implications of this methodology, benefits for the professionals themselves and also the advantages for the institution. It would be interesting if the author could provide some specific data on the results obtained in the implementation of this initiative.

Response: The reviewer reminded me to point out some concepts, which were not clear in the previous version of the manuscript. I gladly accepted the reviewer's valuable advice which helps to make this methodological study more accessible.

I stressed that the manuscript aims to present in detail an initiative in methodology applied in occupational settings. I have placed this sentence at the end of the Introduction.

The reviewer correctly underlines the integrative approach of the implemented method. I thought it useful to say, at the beginning of the Materials and Methods section, that in order to implement this integrative approach it is essential to provide above all for the prevention of occupational risks. Only when this is achieved is it legitimate to plan the promotion of workers' health.

I thought it appropriate, also in the light of what the reviewer asked, to explain that the ASIA method for risk management and the GEP technique for worker participation in job improvement are techniques taught by our school.

The results of the individual promotion campaigns have been the subject of publications that we have mentioned in the text and to which we refer readers who want to know them in detail.

Workers were invited to participate in WHPEMS action during their regular medical examination in the workplace. There is no sampling, because all those undergoing health surveillance are invited to participate. The workers are free to accept or refuse, but the vast majority agree to provide the requested data and are admitted to promotion. For greater clarity, I have added this specification in the third paragraph of the Materials and Methods section. I also explained (in the same paragraph) that the questionnaire form can be provided upon request to doctors who intend to apply this method in their companies and that workers' answers can also be collected online. No charge is required for these services, which are aimed at improving health in the workplace also in other companies.

The cohort affected by the promotion programs is mobile, not only because some workers enter companies for new hires or leave them due to resignations or retirement, but also because companies change the doctor to whom they have given the task of supervising the workers. Prospective data analysis is only possible if the mandate is maintained. In the past I was able to publish some longitudinal studies and I hope this will still be possible in the future. At the moment, I reported on cross-sectional data.

According to reviewers’ request, I added a few lines to explain that In Italy, an incentive for the implementation of health promotion programs is represented by the fact that the companies that carry out interventions supplementary to the obligatory prevention of occupational damage can request a reduction in the insurance premiums they have to pay. This immediate economic benefit largely exceeds the very modest expense of WHPEMS programs, and ensures that any other benefit of the intervention that should occur in the future due to reduced absenteeism, increased productivity, etc., will be a net gain.

In the subsequent paragraph I have reported some benefit for workers. Choosing topics shared with workers ensures that program goals are relevant to workers' health, and that they will derive the greatest benefit for their own health. In the Discussion, I have underlined also the advantages for occupational health physicians. The implementation of WHPEMS interventions is an advantage for the occupational doctor, who has the possibility of improving the health of workers and earning an optimal relationship with them. It is useful to remember that the occupational doctor is not chosen by the workers and does not enjoy that fundamental doctor-patient relationship on which medicine is based. By implementing WHPEMS programs and improving working life and work culture he/she can gain the trust and cooperation of workers.